# Bayesian Sequential Batch Design in Functional Data

## Abstract

Many longitudinal studies are hindered by noisy observations sampled at irregular and sparse time points. In handling such data and optimizing the design of a study, most of the existing functional data analysis focuses on the frequentist approach that bears the uncertainty of model parameter estimation. While the Bayesian approach as an alternative takes into account the uncertainty, little attention has been given to sequential batch designs that enable information update and cost efficiency. To fill the gap, we propose a Bayesian hierarchical model with Gaussian processes which allows us to propose a new form of the utility function based on the Shannon information between posterior predictive distributions. The proposed procedure sequentially identifies optimal designs for new subject batches, opening a new way for incorporating the Bayesian approach in finding the optimal design and enhancing model estimation and the quality of analysis with sparse data.

## 1 Introduction

Many of the longitudinal studies suffer from noisy observations. It is often the case that only a small number of irregularly spaced observations can be taken for each subject, making it a sparse dataset for the subsequent analysis (Zeger and Diggle, 1994; Brumback and Rice, 1998; Guo, 2004; Yao et al., 2005). In light of this issue, functional data analysis (FDA) has been developed as one of the most popular methods to handle such data and enhance the quality of estimation. In particular, as the sparse observations can only provide limited information for recovering the underlying trajectory, FDA offers an effective way to optimize the design of a study by judiciously selecting optimal time points for taking observations.

Existing FDA literature has mostly focused on rather a frequentist approach that considers the "best guess" of parameters to find an optimal design (Ji and Müller, 2017; Park et al., 2018; Rha et al., 2020). However, this approach oftentimes bears uncertainty of the model parameter estimation and can possibly hinder the quality of analysis. A Bayesian approach, on the other hand, takes into account this uncertainty and conducts the analysis based on a prior distribution of the parameters (Chaloner and Verdinelli, 1995). Specifically, a Bayesian hierarchical model assumes a common mean function for the underlying subject trajectories, enabling us to borrow the strength of all observations across subjects to recover the trajectories. (Yang et al., 2016)

Ryan et al. (2015) proposed the fully Bayesian static design for mixed effect model to determine sampling time points for precise estimation of the model parameters. Nevertheless, the static design uses the same design throughout the experimental process without accounting for any incoming information that may be collected during the experiment (Ryan et al., 2016). In this regard, a sequential design may offer more efficient and flexible design schemes as it updates the optimal design at each stage with new information provided from the previous stages. (Chaloner, 1986; Müller et al., 2007). Yet scant work has been done on constructing Bayesian hierarchical models

Submitted to Workshop on Bayesian Decision-making and Uncertainty, 38th Conference on Neural Information Processing Systems (BDU at NeurIPS 2024). Do not distribute.

37 with a sequential design that considers the uncertainty of model parameter estimation and updates the
38 optimal design with newly acquired information at each stage.

39 To fill this gap, in this study, we propose a Bayesian hierarchical model that sequentially identifies
40 optimal designs for new batches of subjects by (1) providing information for updating the posterior
41 mean function of the underlying trajectories of existing subjects and (2) offering sufficient information
42 for accurate estimation of new subject trajectories. Particularly, we first obtain the posterior distribu-
43 tions of underlying trajectories from our Bayesian hierarchical model, and update the distributions
44 with new observations. Then based on the posterior distributions, we find the optimal design by the
45 simulated annealing (SA) algorithm proposed by Van Laarhoven et al. (1987), which is widely known
46 for its strengths in search in large space and computational efficiency.

47 In sum, our study is expected to open a new way for incorporating the Bayesian approach in handling
48 noisy observations with sequential batch designs and further enhance model estimations with new
49 information update. The rest of paper is organized as follows: Section 2 introduces our Bayesian
50 hierarchical model that is used to obtain the posterior distributions of underlying trajectories. Section
51 3 formulates the utility function as the design criterion for finding the optimal design. Section 4
52 details the implementation of the simulated annealing algorithm on the search of the optimal design.
53 A discussion can be found in Section 5.

54 ## 2 Bayesian Hierarchical Model

55 In longitudinal studies, it is not uncommon to have observations that are sampled at sparse and
56 irregular time points. The collected samples are viewed as functional observations and are often
57 contaminated with unknown noises. Assuming each subject following their independent stochastic
58 process, we consider the Bayesian hierarchical model proposed by Yang et al. (2016) as follows:

$$\boldsymbol{Y}_i(\boldsymbol{t}_i) = \boldsymbol{X}_i(\boldsymbol{t}_i) + \boldsymbol{\epsilon}_i, \quad \boldsymbol{\epsilon}_i \overset{i.i.d.}{\sim} N(\boldsymbol{0}, \sigma_\epsilon^2 \boldsymbol{I}),$$

$$\boldsymbol{X}_i \mid \boldsymbol{\mu}, \boldsymbol{\Sigma} \overset{i.i.d.}{\sim} GP(\boldsymbol{\mu}, \boldsymbol{\Sigma}), \qquad i = 1, \ldots, n,$$

$$\boldsymbol{\mu} \sim GP\left(\boldsymbol{\mu_0}, \frac{1}{c}\boldsymbol{\Sigma}\right),$$

59 where $\boldsymbol{Y}_i(\boldsymbol{t}_i) = \{Y_i(t_{i,1}), \ldots, Y_i(t_{i,n_i})\}$ are the noisy observations of the underlying trajectory $\boldsymbol{X}_i$
60 at time $\boldsymbol{t}_i = (t_{i,1}, \ldots, t_{i,n_i})'$. We consider the additive error vector $\boldsymbol{\epsilon}_i$ that follows i.i.d. normal
61 with mean vector $\boldsymbol{0}$ and variance $\sigma_\epsilon^2 \boldsymbol{I}$ and is independent of $\boldsymbol{X}_i$. We assume each $\boldsymbol{X}_i$ follows i.i.d.
62 Gaussian process with a prespecified mean function $\boldsymbol{\mu}$ and covariance kernel $\boldsymbol{\Sigma}$. The universal mean
63 function $\boldsymbol{\mu}$ is assumed unknown and is assigned with a Gaussian process as $\boldsymbol{\mu} \sim GP(\boldsymbol{\mu}_0, (1/c)\boldsymbol{\Sigma})$
64 with the mean function $\boldsymbol{\mu}_0$ and the covariance kernel $\boldsymbol{\Sigma}$ scaled by some $c > 0$. For simplicity, we
65 denote $\boldsymbol{Y}_i(\boldsymbol{t}_i)$ by $\boldsymbol{Y}_{i,t_i}$, $\boldsymbol{X}_i(\boldsymbol{t}_i)$ by $\boldsymbol{X}_{i,t_i}$, $\boldsymbol{\mu}(\boldsymbol{t}_i)$ by $\boldsymbol{\mu}_{t_i}$, and $\boldsymbol{\Sigma}(\boldsymbol{t}_i, \boldsymbol{t}_i)$ by $\boldsymbol{\Sigma}_{t_i,t_i}$. Given time grid $\{\boldsymbol{t}_i\}$,
66 we have the following hierarchical structure in multivariate forms for subject $i$:

$$\boldsymbol{Y}_{i,t_i} | \boldsymbol{X}_{i,t_i} \sim MVN(\boldsymbol{X}_{i,t_i}, \sigma_\epsilon^2 \boldsymbol{I}),$$

$$\boldsymbol{X}_{i,t_i} | \boldsymbol{\mu}_{t_i}, \boldsymbol{\Sigma}_{t_i,t_i} \sim MVN(\boldsymbol{\mu}_{t_i}, \boldsymbol{\Sigma}_{t_i,t_i}),$$

$$\boldsymbol{\mu}_{t_i} | \boldsymbol{\mu}_0, \boldsymbol{\Sigma} \sim MVN(\boldsymbol{\mu}_{0t_i}, \frac{1}{c}\boldsymbol{\Sigma}_{t_i,t_i}). \tag{1}$$

67 For simplicity, we assume that the error variance is fixed and the covariance kernel to follow a
68 pre-specified structure as squared exponential kernel. The scaling constant $c$ for the covariance kernel
69 of the mean function is set to 1 and thus does not require posterior update in the estimation step. For
70 the hyperparameter $\boldsymbol{\mu}_0$, We set it to be the smoothed sample mean of $\{\boldsymbol{Y}_{i,t_i}\}$.

71 Different from previous approaches in functional data analysis that mainly focus on smoothing each
72 curve individually, the hierarchical GP model borrows the strength of all observations and smooth
73 the entire functional observations at once by assuming a common mean function $\boldsymbol{\mu}$ (Yang et al.,
74 2016). In addition, two layers of GPs with the same covariance kernel function provide important
75 insights and computational efficiency to our design problem. Assigning a GP on $\boldsymbol{\mu}$ allows the model
76 to share information across the subjects and to predict the trajectories at unobserved time grids for
77 all of the subjects based on the collected observations of only a portion of subjects. Besides, the
78 hierarchical structure of GPs still gives us a closed form of the predictive distribution which reduces
79 the computational cost in evaluating the optimal design criterion significantly. In the next section, we
80 will detail the design problem and propose a utility function for the corresponding optimal design.

## 3 Utility Function and Optimal Sequential Batch Design

Conventional sequential design approach adopts one-step-look-ahead method that only considers the next subject, which is often not optimal. Static design approach determines the optimal design in a holistic view but uses the same fixed protocol throughout the experiments. To combine the best of two worlds, we adopt a sequential batch scheme. We consider the problem of multistage design that sequentially finds optimal sampling times for a new batch of subjects based on the information obtained from observations of existing subjects from previous stages. For demonstration purposes, we only display the utility function for one future stage. However, by including new observations with the obtained optimal design at the current stage, one is able to update the optimal design criterion and acquire new optimal designs for all future stages in a sequential manner.

For the experiments, we assume that observations can be taken on an equally-spaced common grid that has $T_0$ time points. Yet, for each subject, only $k(< T_0)$ observations can be taken. Before stage 1, we assume that an experiment is already conducted and observations $\boldsymbol{Y}_0 = \{\boldsymbol{Y}_1(\boldsymbol{t}_i), \ldots, \boldsymbol{Y}_N(\boldsymbol{t}_i)\}$ for $N$ subjects are taken based on a fixed design $\boldsymbol{D}_0 = \{\boldsymbol{t}_1, \ldots, \boldsymbol{t}_N\}$. Suppose we are now at stage 1 and we are to recruit a new batch of $M(> 1)$ subjects and take observations $\boldsymbol{Y}_1 = \{\boldsymbol{Y}_{N+1}(\boldsymbol{t}_i), \ldots, \boldsymbol{Y}_{N+M}(\boldsymbol{t}_i)\}$ from these subjects according to a design $\boldsymbol{D}_1 = \{\boldsymbol{t}_{N+1}, \ldots, \boldsymbol{t}_{N+M}\}$. Here, we consider the batch size $M$ and the number of observations per subject $k$ to be fixed. Our attempt is to find the optimal design $\boldsymbol{D}_1$ that achieves two goals: (1) the newly-added observations based on $\boldsymbol{D}_1$ should provide more information to update the posterior mean function so as to improve the recovery of underlying trajectories $\boldsymbol{X}_0$ for the existing subjects $1, \ldots, N$; (2) the observations based on $\boldsymbol{D}_1$ should also provide sufficient information for the estimation of new batch of subject trajectories.

Specifically, when recovering the trajectories of existing and new subjects, we focus on the trajectory values at unobserved time points, denoted by $\boldsymbol{X}^c$. We would like to compare the posterior predictive distributions $p(\boldsymbol{X}_0^c, \boldsymbol{X}_1^c|\boldsymbol{Y}_0)$ of $\boldsymbol{X}_0^c$ and $\boldsymbol{X}_1^c$ given the information from existing subjects to the posterior predictive distributions $p(\boldsymbol{X}_0^c, \boldsymbol{X}_1^c|\boldsymbol{Y}_0, \boldsymbol{Y}_{1,\boldsymbol{D}_1})$ of $\boldsymbol{X}_0^c$ and $\boldsymbol{X}_1^c$ given the information from existing subjects and the new batch of subjects. That is, we would like to maximize the improvement in prediction of $\boldsymbol{X}^c$ before and after including the new batch of subjects.

We consider an information-based approach and measure the improvement by Kullback-Leibler (KL) divergence, which is a classic metric in information theory that measures the difference between two distributions. Therefore, we propose the following utility function as the optimal design criterion:

$$U(\boldsymbol{D}_1, \boldsymbol{Y}_0) = D_{KL}(p_1||p_0) = \int \log\left(\frac{p_1}{p_0}\right) dp_1, \tag{2}$$

where we denote by $p_0 = p(\boldsymbol{X}_0^c, \boldsymbol{X}_1^c|\boldsymbol{Y}_0)$ and $p_1 = p(\boldsymbol{X}_0^c, \boldsymbol{X}_1^c|\boldsymbol{Y}_0, \boldsymbol{Y}_{1,\boldsymbol{D}_1})$, which are both multivariate normal distributions under our model framework.

To evaluate the above utility function, we consider a combination of implementing the predictive formula of Gaussian process and using empirical Bayes procedure for the rest of model parameters to obtain a closed-form solution for the utility function. Concerning the page limit, we refer the readers to Appendix A for the detailed derivation. This closed-form solution facilitates computational efficiency by avoiding the evaluation of intractable marginal likelihood in the utility function as commonly seen in many optimal Bayesian design problems.

## 4 Computation

Because of the closed-form solution of the utility function in Section 3, it is easy to evaluate the utility function with a given design. Yet, the design space remains large as we are exploring optimal designs for a batch of subjects simultaneously. Therefore, we implement a simulated annealing (SA) algorithm (Van Laarhoven et al., 1987) that enables efficient exploration of large and complex design spaces and easy implementation. Specifically, the SA algorithm is used at every stage such that it incorporates existing and new information from all previous and current stages and finds optimal design for the next stage in a sequential manner.

The SA algorithm starts with an initial "temperature" $T_{initial}$ and a randomly generated design $\boldsymbol{D}_{initial}$. The "energy" $e$ of this design is then computed based on the utility function defined in Equation (2). Then the algorithm generates another candidate design $\boldsymbol{D}_{test}$ from the "neighborhood"

of $\boldsymbol{D}_{initial}$ and calculates its energy $e_{test}$. If the difference between two energies $\Delta e = e - e_{test} \leq 0$, the candidate design $\boldsymbol{D}_{test}$ is accepted and the algorithm will continue to compare it to other neighborhood designs. At the current temperature $T$, if $\Delta e > 0$, the candidate design is accepted with a probability of $\exp\left(\Delta e / T\right)$. This process is repeated until no further improvements can be made within a maximum number of iterations. Then the temperature will be lowered according to a "cooling schedule" and the whole procedure will be repeated again. Finally, we follow the approach proposed by Aragon et al. (1991) to terminate the algorithm if the acceptance probability is smaller than some threshold $P_{threshold}$.

In the algorithm, a number of parameters, initial temperature, cooling schedule, neighborhood of a design, maximum number of iterations, and acceptance threshold, require initial values. Nevertheless, as the SA algorithm is a heuristic algorithm, the parameter values heavily depend on the problem settings and experiment setup. Therefore, we also set the parameter values in a heuristic way so as to be able to adapt to different scenarios. Based on suggestions in Van Laarhoven et al. (1987), we set the initial temperature $T_{initial}$ to be $\Delta e / \log(0.7)$ so that the initial acceptance probability for designs with $\Delta e > 0$ is 0.6. This is to limit the time spent at high temperatures. The cooling schedule is an exponential decaying function of the temperature $T_{new} = 0.95 \times T_{old}$.

For the neighborhood of a design, there are many choices, such as changing only one time point for one subject in the batch or changing one set of time points for one subject in the batch. However, the candidate set for the former can easily increase exponentially with different time grid and observation sizes and it is also suspected that a single time point can make much difference on the trajectory recovery of all subjects. Thus, considering computation efficiency, we define the neighborhood of a design by changing one set time points from one subject in the batch. Here we propose to set the maximum number of iterations to be 10 and the acceptance threshold to be 0.2, as suggested in Aragon et al. (1991). As noted before, since the SA algorithm is a heuristic approach that is contingent upon a specific problem, empirical tuning on the initial parameters is necessary when conducting different experiments. A pseudo code that illustrates the structure of the algorithm can be found in Appendix B.

# 5   Discussion

To handle the noisy observations in many fields such as longitudinal studies, extant FDA literature mostly adopts rather a frequentist approach and bears the uncertainty of parameter estimation. As an alternative to improve the quality of model estimation, a Bayesian approach naturally takes into account the uncertainty in estimation and produces posterior predictive distribution. In this study, we adopt a Bayesian hierarchical model of Gaussian processes for the underlying trajectories, which enables us to obtain the trajectory predictive distributions with closed-form expressions at reduced computational cost. We propose an optimal Bayesian sequential batch design scheme that sequentially finds optimal design for a batch of subjects based on the information obtained from all previous and current stages. Specifically, its sequential feature helps update the optimal design criterion with new information at each stage, whereas its batch feature controls for a small number of stages and maintains the overall cost effectiveness. Combining these two features, this scheme is designed to improve the trajectory recovery of current subjects and achieve accurate estimation of future subject trajectories. Finally, in the optimization step, we implement a simulated annealing algorithm that takes in empirically-tuned parameters and outputs a final design with computational efficiency.

Further refinement of this study can be done by altering the assumptions made in our analysis. Particularly in the design setup, we assume that the batch size $M$ of the optimal design is small. This is established as $M$ should not be too large to only have too few updates on the design optimality criterion. Nonetheless, in practice, $M$ is often contingent upon the size of the initial data set and the number of design stages. The interactions between these factors may change the optimal size of the batch. To account for this, there are two potential approaches to find the optimal $M$. One is to iteratively test different values of $M$ from 1 to the existing subject size $N$. Yet additional consideration will need to be put in to reduce its computational expensiveness. Another is to include $M$ as a random variable and incorporate it inside the utility function. That is, the optimal design and the optimal batch size are obtained in each stage.

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

## A Derivation of Utility Function

Let $A$ be $(\boldsymbol{X}_0^c, \boldsymbol{X}_1^c)|\boldsymbol{Y}_0$ with distribution $p_0$ and let $B$ be $(\boldsymbol{X}_0^c, \boldsymbol{X}_1^c)|(\boldsymbol{Y}_0, \boldsymbol{Y}_{1,\boldsymbol{D}_1})$ with distribution $p_1$. We first derive the distribution $p_1$ of $B$, then the distribution $p_0$ of $A$ follows by omitting $\boldsymbol{Y}_{1,\boldsymbol{D}_1}$. For notation simplicity, let $\boldsymbol{Y}_B$ be $(N + M) \times k$ dimensional vector containing the observations from existing and new batch of subjects, let $\boldsymbol{X}^c$ be $(N + M) \times (T_0 - k)$ dimensional vector containing the underlying trajectory values evaluated at unobserved time points. And let $\boldsymbol{t}$ be the time points that have observations for subjects $1, \ldots, N + M$, and let $\boldsymbol{t}^c$ be the time points that have missing values for subjects $1, \ldots, N + M$.

Recall in the Bayesian hierarchical model (1) in Section 2, we assume multivariate distributions for the finite observations and underlying trajectory values. We may obtain the joint distribution of $\boldsymbol{Y}_B$ and $\boldsymbol{X}_c$ given the hyperparameter $\boldsymbol{\mu}_0$ as follows:

$$\begin{pmatrix} \boldsymbol{Y}_B \\ \boldsymbol{X}_c \end{pmatrix} \bigg| \boldsymbol{\mu}_0 \sim MVN\left( \begin{pmatrix} \boldsymbol{\mu}_0(\boldsymbol{t}) \\ \boldsymbol{\mu}_0(\boldsymbol{t}^c) \end{pmatrix}, \begin{pmatrix} (1+c)\boldsymbol{\Sigma}(\boldsymbol{t},\boldsymbol{t}) + \sigma_\epsilon^2 \boldsymbol{I} & \boldsymbol{\Sigma}(\boldsymbol{t},\boldsymbol{t}^c) \\ \boldsymbol{\Sigma}(\boldsymbol{t}^c,\boldsymbol{t}) & (1+c)\boldsymbol{\Sigma}(\boldsymbol{t}^c,\boldsymbol{t}^c) \end{pmatrix} \right),$$

$$\text{where } \boldsymbol{X}^c = \begin{pmatrix} \boldsymbol{X}_0^c \\ \boldsymbol{X}_1^c \end{pmatrix}, \boldsymbol{Y}_B = \begin{pmatrix} \boldsymbol{Y}_{0,\boldsymbol{D}_0} \\ \boldsymbol{Y}_{1,\boldsymbol{D}_1} \end{pmatrix}.$$

Then with the joint distribution, we may derive the conditional distribution of $\boldsymbol{X}_c|\boldsymbol{Y}_B$ by the conditional expectation property of multivariate normal distribution. Therefore, we get the distribution of B as

$$B = \boldsymbol{X}_c|\boldsymbol{Y}_B \sim MVN(\boldsymbol{m}_B, \boldsymbol{\nu}_B),$$

$$\text{where } \boldsymbol{m}_B = \boldsymbol{\mu}_0(\boldsymbol{t}) + \boldsymbol{\Sigma}(\boldsymbol{t},\boldsymbol{t}^c)((1+c)\boldsymbol{\Sigma}(\boldsymbol{t}^c,\boldsymbol{t}^c))^{-1}(\boldsymbol{y} - \boldsymbol{\mu}_0(\boldsymbol{t}^c)),$$

$$\boldsymbol{\nu}_B = ((1+c)\boldsymbol{\Sigma}(\boldsymbol{t},\boldsymbol{t}) + \sigma_\epsilon^2 \boldsymbol{I}) - \boldsymbol{\Sigma}(\boldsymbol{t},\boldsymbol{t}^c)((1+c)\boldsymbol{\Sigma}(\boldsymbol{t}^c,\boldsymbol{t}^c))^{-1}\boldsymbol{\Sigma}(\boldsymbol{t}^c,\boldsymbol{t}).$$

One thing worth noting is that the error variance $\sigma_\epsilon^2$ is unknown. To keep computation simplicity, we adopt the empirical Bayes method that uses the maximum likelihood estimator $\hat{\sigma}_\epsilon^2$ as the estimated value of $\sigma_\epsilon^2$.

After getting the distribution of $B$, we may obtain the distribution of $A$ by letting $\boldsymbol{Y}_A = (\boldsymbol{Y}_{0,\boldsymbol{D}_0})$ to be $N \times k$ dimensional vector. Then we substitute $\boldsymbol{Y}_B$ with $\boldsymbol{Y}_A$ and obtain the joint distribution of $\boldsymbol{Y}_A$ and $\boldsymbol{X}_c$ given the hyperparameter $\boldsymbol{\mu}_0$ as:

$$\begin{pmatrix} \boldsymbol{Y}_A \\ \boldsymbol{X}_c \end{pmatrix} \bigg| \boldsymbol{\mu}_0 \sim MVN\left( \begin{pmatrix} \boldsymbol{\mu}_0(\boldsymbol{t}) \\ \boldsymbol{\mu}_0(\boldsymbol{t}^c) \end{pmatrix}, \begin{pmatrix} (1+c)\boldsymbol{\Sigma}(\boldsymbol{t},\boldsymbol{t}) + \sigma_\epsilon^2 \boldsymbol{I} & \boldsymbol{\Sigma}(\boldsymbol{t},\boldsymbol{t}^c) \\ \boldsymbol{\Sigma}(\boldsymbol{t}^c,\boldsymbol{t}) & (1+c)\boldsymbol{\Sigma}(\boldsymbol{t}^c,\boldsymbol{t}^c) \end{pmatrix} \right),$$

$$\text{where } \boldsymbol{X}^c = \begin{pmatrix} \boldsymbol{X}_0^c \\ \boldsymbol{X}_1^c \end{pmatrix}, \boldsymbol{Y}_A = (\boldsymbol{Y}_{0,\boldsymbol{D}_0}).$$

Similarly, by the conditional expectation property of multivariate normal distribution, we get the distribution of $A$ as

$$A = \boldsymbol{X}_c|\boldsymbol{Y}_A \sim MVN(\boldsymbol{m}_A, \boldsymbol{\nu}_A),$$

$$\text{where } \boldsymbol{m}_A = \boldsymbol{\mu}_0(\boldsymbol{t}) + \boldsymbol{\Sigma}(\boldsymbol{t},\boldsymbol{t}^c)((1+c)\boldsymbol{\Sigma}(\boldsymbol{t}^c,\boldsymbol{t}^c))^{-1}(\boldsymbol{y} - \boldsymbol{\mu}_0(\boldsymbol{t}^c)),$$

$$\boldsymbol{\nu}_A = ((1+c)\boldsymbol{\Sigma}(\boldsymbol{t},\boldsymbol{t}) + \sigma_\epsilon^2 \boldsymbol{I}) - \boldsymbol{\Sigma}(\boldsymbol{t},\boldsymbol{t}^c)((1+c)\boldsymbol{\Sigma}(\boldsymbol{t}^c,\boldsymbol{t}^c))^{-1}\boldsymbol{\Sigma}(\boldsymbol{t}^c,\boldsymbol{t}).$$

Lastly, since both $A$ and $B$ follow multivariate normal distributions, the closed-form of the KL divergence between two multivariate normal distributions is

$$D_{KL}(p_1||p_0) = \frac{1}{2}\left[ \log\left( \frac{|\boldsymbol{\nu}_A|}{|\boldsymbol{\nu}_B|} \right) - k + tr\left\{ \boldsymbol{\nu}_A^{-1}\boldsymbol{\nu}_B \right\} + (\boldsymbol{m}_A - \boldsymbol{m}_B)^T \boldsymbol{\nu}_A^{-1}(\boldsymbol{m}_A - \boldsymbol{m}_B) \right].$$

 # B Pseudo Code for Simulated Annealing Algorithm

---

**Algorithm 1** Simulated-Annealing Algorithm

---

$\boldsymbol{D} \leftarrow \boldsymbol{D}_{initial}$
$e \leftarrow Energy(\boldsymbol{D}_{initial})$
$T \leftarrow T_{initial}$
**while** $\exp{(\Delta e/T)} > 0.2$ **do**
    $\boldsymbol{D}_{test} \leftarrow neighborhood(\boldsymbol{D}_{initial})$
    $e_{test} = Energy(\boldsymbol{D}_{test})$
    $\Delta e = e - e_{test}$
    **if** $\Delta e \leq 0$ **then**
        $\boldsymbol{D} \leftarrow \boldsymbol{D}_{test}$
        $e \leftarrow e_{test}$
    **else**
        $q \leftarrow Random(0, 1)$
        **if** $q < \exp{(\Delta e/T)}$ **then**
            $\boldsymbol{D} \leftarrow \boldsymbol{D}_{test}$
            $e \leftarrow e_{test}$
        **end if**
    **end if**
    $T = 0.95 \times T$
**end while**

---

