# OpenReview forum: "Bayesian Sequential Batch Design in Functional Data"
_NeurIPS.cc/2024/Workshop/BDU — Submitted to NeurIPS BDU Workshop 2024_

### Official Review · Reviewer_9HHH · 2024-09-20
**Limited Contribution**

**Rating:** 3
**Confidence:** 4

**Review:**

In this work, the authors explore the problem of optimal sequential batch design for functional data using a previously introduced model (https://projecteuclid.org/journals/bayesian-analysis/volume-11/issue-3/Smoothing-and-MeanCovariance-Estimation-of-Functional-Data-with-a-Bayesian/10.1214/15-BA967.full) . Two main contributions presented is the defining of a KL based decision criterion and a simulated annealing based algorithm to estimate said criterion.

Unfortunately, neither seem to present substantial contributions. The former is a straight forward exercise in computing the KL divergence of two gaussian processes and the latter utilizes simulated annealing as their approach. While the authors claim that simulated annealing is "enables efficient exploration of large and complex design spaces" citing a 1987 book on the matter, more recent works such as (https://www.jstor.org/stable/44162464) ,which was cited in this paper, note a variety of studies which show the opposite. That simulated annealing actually has poor convergence properties in large and complex spaces. This makes this particularly ill-suited for this optimal design problem.

The paper also has no simulation or real-world data section so it is difficult to determine if the proposed approach is practical work works well in practice.

---

### Official Review · Reviewer_NTtz · 2024-09-27
**The usefulness of the method is not clear to me**

**Rating:** 4
**Confidence:** 4

**Review:**

Pros:
* Solves a problem not previously solved, as far as I know. There is not a lot of work on active learning or Bayesian Optimization with time-series data that has been measured at sparse and irregular points in time.
* Clear exposition of the method in Sections 2 and 3

Cons:
* Unclear where this method would be useful
* Intermittently confusing writing
* Limited engagement with the literatures on Bayesian adaptive experimental design
and active learning; I suspect there is a better implementation
* The method does not appear to have been implemented; seeing that it gives reasonable results on even a small toy example would have been reassuring. The devil can be in the details.

*Contribution*: The authors suggest a method for choosing at which times to collect data from each
new batch of subjects in a study. In this setting, each subject will be observed a limited number times,
and their entire trajectory over a period of time is of interest, as are the trajectories of other subjects. The method chooses observation
times in order to maximize information learned about the trajectories of both current
and past subjects, using a hierarchical model that shares strength between subjects.
Although the authors describe this as an experiment, there is no treatment whose effect is
being measured or optimized, so it might be clearer to call it an active learning problem. Although
I am not sure about the usefulness of the model and action space used here, the general setting of functional data (or time-series data more specifically) with sparse observations and active learning seems potentially quite interesting.

I see a few issues with this paper.

First, it was not clear to me what the use cases are. Mentioning even one would have greatly strengthened the paper. I am aware of many cases of sparse
longituidinal data; it is common with human subjects. However,
in these cases, the researcher may have limited control over the data collection schedule.
Subjects may drop out or become temporarily unreachable. Data collection may be geographically
staggered, with subjects who move presenting special challenges. I may be envisioning the wrong
sort of application here, since the authors didn't mention one. The setup in this paper is further
limited by the fact that data collection ends for one batch of subjects before it begins
for the next. It is slightly odd that despite this being an adaptive setting, there is no adaptiveness over time within a batch,
but that might be a reasonable simplification. I am not sure if there is a motivation for this very specific setup.

*Originality*: On the plus side, I do believe the problem being solved is original, as far as I know.
There is a rich literature on Bayesian adaptive experimental design (see e.g. the
book by Berry, Carlin, Lee, and Muller), but I'm not aware of whether it addresses the choice of measurement times or how deeply
it considers the challenges associated with functional data.

*Clarity*: I found the writing sometimes clear and sometimes difficult to follow. While sections 2 and 3,
explaining the method, were clear, the introduction and conclusion confused me.
For example, what does it mean that frequentist methods "bear the uncertainty" of
parameter estimation, a claim repeatedly made? (Since this paper uses Empirical Bayes, it may technically be a frequentist method,
if the GP parameters are estimated with MLE.) Section 4, explaining simulated
annealing, was mostly unnecessary.


Some more assorted comments:
* Simulated annealing seems like a questionable choice, since the utility function to be optimized
has a closed form. A first-order or second-order method would likely perform better, with random restarts
if the function is not convex.
* I find it odd that the model uses the same covariance function (or a scaled version of it)
for both mu and x|mu. Without a specific application in mind, it's very hard to say whether
this is reasonable.
* Finally, I'd suggest the authors look more into the literature on information-theoretic active learning; it may be possible
to optimize a design with multi-step lookahead in a computationally tractable way.
* I would love to see comparisions to simpler approaches. What do we gain by choosing the measurement points adaptively, rather than with a fixed schedule? When is it useful?

---

### Decision · Program_Chairs · 2024-10-09

**Decision:**

Reject

**Comment:**

Reviews for this work are fairly critical, raising a number of technical issues, in particular involving the use of simulated annealing and other factors. I encourage the authors to carefully examine the reviews, improve on the technical issues mentioned, and resubmit to a future workshop.